# Variation of miRNA Content in Cow Raw Milk Depending on the Dairy Production System

**DOI:** 10.3390/ijms231911681

**Published:** 2022-10-02

**Authors:** Loubna Abou el qassim, Sandrine Le Guillou, Luis J. Royo

**Affiliations:** 1Servicio Regional de Investigación y Desarrollo Agroalimentario (SERIDA), 33300 Villaviciosa, Spain; 2GABI, INRAE, AgroParisTech, Université Paris-Saclay, 78350 Jouy-en-Josas, France; 3Department of Functional Biology, Genetics, University of Oviedo, 33006 Oviedo, Spain

**Keywords:** milk, microRNA, biomarker, dairy production systems

## Abstract

Pasture-based milk presents several advantages over milk from intensive industrial farming in terms of human health, the environment, animal welfare, and social aspects. This highlights the need for reliable methods to differentiate milk according to its origin on the market. Here, we explored whether miRNA profiles could serve as a marker of milk production systems. We compared levels of previously described miRNAs in milk from four production systems (altogether 112 milk samples): grazing, zero grazing, grass silage or corn silage. Total RNA was extracted from the fat phase, and miRNAs levels were quantified by real-time quantitative PCR. The levels of the miRNAs *bta-miR-155* and *bta-miR-103* were higher in the grazing system than in corn silage farms. The levels of *bta-miR-532*, *bta-miR-103* and *bta-miR-7863* showed differences between different farm managements. The miRNAs *bta-miR-155* and *bta-miR-103* were predicted to participate in common functions related to fat metabolism and fatty acid elongation. All four differentially expressed miRNAs were predicted to participate in transport, cell differentiation, and metabolism. These results suggest that the dairy production system influences the levels of some miRNAs in milk fat, and that *bta-miR-155* and *bta-miR-103* may be potential biomarkers to identify milk from pasture-managed systems.

## 1. Introduction

Milk production systems vary in how extensive or intensive they are, ranging from one extreme of extensive pastoral livestock farming to the other extreme of intensive industrial farming [1]. In pasture-based production systems, animals with a genotype appropriate to the area feed on pastures or rangelands, external feed inputs are minimized, the use of pastoral resources is optimized, and the stocking rate is low [1,2,3,4]. At the other extreme, intensive farms rely on a high stocking rate and other measures to maximize milk production per cow. The animals are permanently housed and eat a diet based on silage and large amounts of concentrates [5]. Many farms combine certain characteristics between these two extremes, where animals may graze, receive supplements with conserved forages and concentrates and, depending on the climatic conditions, be stabled or on pastures [4,6].

Pasture-based milk production has environmental benefits over intensive production [2,7], as the milk contains higher levels of functional nutrients [8,9] and the marginal milk cost may be lower [10,11,12]. Pasture-based milk production is also more animal friendly [13] and it reduces workload, improving farmer lifestyle and creating a positive image of livestock farming [4].

Except for some regional or sectoral initiatives, no national regulations clearly define the differences among this continuum of milk production systems [1]. This makes it difficult for more or less extensive farms to certify the advantages of their milk to consumers who demand socially and environmentally responsible products [14,15]. A traceability system to identify the milk production system based on markers in the milk itself could support the certification and accurate marketing of milk from less intensive production systems.

The present study explored whether levels of microRNAs (miRNAs) in milk might serve as indicators of how extensively or intensively it was produced. As small, endogenous non-coding RNAs of 21–25 nucleotides, miRNAs bind to specific targets in mRNA to regulate their expression and thereby control various processes within cells [16]. Beyond their functions in the cells that produce them, microRNAs can also be transferred to other cells, or to other species, in protein complexes or through extracellular vesicles [17,18]. There is also compelling evidence that humans use microRNAs from cow’s milk in gene regulation [19,20], highlighting the bioactive characteristic of milk.

Various body fluids contain miRNAs, including tears, colostrum, plasma, and seminal fluids [21] as well as milk [22]. Milk miRNAs have already shown potential as biomarkers of mammary gland diseases [23,24], the cow’s physiological state [25] and stress [26], diet [27,28,29], and breed [30]. Such miRNAs can also serve as a quality assurance indicator to verify labeling on milk powder [31].

Given that miRNAs expression varies according to cow genotype [30,32] and environment [26,33], we wondered whether miRNAs profiles might differ reliably among milk samples from different production systems. In support of this idea, gene expression in cow mammary glands has been shown to depend on diet [34], exercise [35], and stress [36]. If so, milk miRNAs could be an ideal biomarker of the production system, given that miRNAs are stable to high temperature, freeze/thaw cycles, RNase digestion and low pH [37]. In addition, they can be sampled in a non-invasive or minimally invasive manner [38]. Hence, in the current study, the quantification of the abundance of 12 selected miRNAs was performed in tank milk from four dairy farming systems: grazing, zero grazing, grass silage or corn silage. The miRNAs were selected from a previous sequencing work [39], and others were selected from the literature for being associated with feeding and metabolism. By comparing the abundance levels of these miRNAs, we aim to highlight potential non-invasive biomarkers of the milk production systems, which may contribute to the authentication of socially and environmentally responsible dairy products.

## 2. Results

### 2.1. MiRNAs with Differential Levels in Cow Milk According to Production System

Total RNA concentrations in milk fat from the four types of milk production system varied between 84 and 144 ng/µL, and the RNA was of good quality: the absorbance ratio was 1.67–1.98 in all samples.

MiRNA levels in milk fat across the four dairy production systems for the 12 chosen miRNAs were estimated. We found that levels of the following four miRNAs differed significantly between at least two dairy production systems: *bta-miR-155*, *bta-miR-103*, *bta-miR-532*, and *bta-miR-7863* (Figure 1). Post hoc analysis showed that miRNAs *bta-miR-103* and *bta-miR-155* showed significant differences between the grazing and corn silage groups, being more abundant in the grazing farms. The *bta-miR-532* was significantly more abundant in grazing farms than in zero grazing, while on the contrary *bta-miR-7863* was more abundant in zero-grazing farms than in grazing. *Bta-miR-103*, *bta-miR-532* and *bta-miR-7863* showed significant differences between zero grazing and corn silage groups, with *bta-miR-103* and *bta-miR-7863* being more abundant in the zero-grazing group, but *bta-miR-532* was more abundant in the corn silage group. The miRNA *bta-miR-532* was significantly more abundant in corn silage farms than in grass silage (Figure 1).

### 2.2. miRNA Functionality and Pathway Analyses

To determine the possible implications of the studied miRNAs in the biological response to different production systems, we predicted the target genes of the four miRNAs as well as the functional pathways in which those target genes may participate. The 707 targets of *bta-miR-103* were associated with 71 KEGG pathways (The Kyoto Encyclopedia of Genes and Genomes), 8 biological processes, and 15 molecular functions. The 460 targets of *bta-miR-155* were associated with 106 KEGG pathways, 5 biological processes, and 9 molecular functions. The 2266 targets of *bta-miR-7863* were associated with 76 KEGG pathways, 12 biological processes, and 7 molecular functions. The 208 targets of *bta-miR-532* were associated with 29 KEGG pathways, 3 biological processes, and 22 molecular functions.

Among these target processes, we identified 15 KEGG pathways (Table 1), 8 biological processes (Table 2), and 12 molecular functions (Table 3) that were related to milk production and metabolism. In particular, the four miRNAs were all predicted to be involved in the MAPK signaling pathway (mitogen-activated protein kinases signaling pathway) and the molecular functions of transferases and serine/threonine-protein kinases. Two metabolic pathways stand out for their relationship with milk production and secretion, the oxytocin and prolactin signaling pathways.

The miRNAs *bta-miR-103*, *bta-miR-155* and *bta-miR-532* showed high homology to human miRNAs, so we used DIANA miRPath for a second functionality analysis. Using Tarbase, we found that human target genes were experimentally validated for *miR-103a-3p* (2156 genes), *miR-155* (1117 genes), and *miR-532* (306 genes). These targets (using the option union for KEGG and intersection for GO (gene ontology) to merge results were associated with 34 KEGG pathways (Figure 2) and 62 GO categories (Appendix A). These analyses allowed us to predict pathways regulated by the three miRNAs (available in the Tarbase database) (Figure 2 and Appendix A). These results predicted *miR-103a-3p* and *miR-155* to be involved in fatty acid elongation and metabolism.

GO analysis revealed nine categories common to *miR-103a-3p*, *miR-155*, and *miR-532*: protein binding transcription factor activity, cellular protein modification processes, catabolic processes, biosynthetic processes, cellular nitrogen compound metabolic processes, cellular protein metabolic processes, small-molecule metabolic processes, mRNA metabolic processes, generation of precursor metabolites and energy, and regulation of glucose transport.

## 3. Discussion

This work aimed to identify miRNAs that vary according to milk production system in order to differentiate milk from farms managed extensively. In the present study, among the 12 miRNAs analyzed, the levels of 4 miRNAs differed significantly among the types of dairy production system. These findings suggest the potential of profiling miRNAs in milk in order to certify whether it came from an extensive grazing production system.

Previous work from our group showed that the levels of the miRNA *bta-miR-215* in milk fat differed between the two extremes of extensiveness/intensiveness [40]. The present work is not limited to extreme production systems but compares grazing livestock farms with intensive and mixed managements.

The levels of *bta-miR-103* were highest in farms including fresh grass in the diet, either grazed or harvested. Previous work showed that the expression of *bta-miR-103* in blood [41] and subcutaneous fat [42] was similar between cows fed on pasture and those fed in a free-stall barn with fresh grass harvested every morning. The fresh grass delivery mode does not affect the expression of *bta-miR-103*.

In a study aiming to evaluate the effect of pasture during 3 months on stearoyl-CoA desaturase (SCD) and *miR-103* expression in milk dairy goats, the SCD was significantly higher in grazing animals compared to housed animals consuming conserved forages, whereas *miR-103* tended to be higher but not significantly [43]. Additionally, Lin et al., (2013) [44] showed that *miR-103* and SCD gene expression had similar trends, and that the overexpression of *miR-103* in mammary gland has been linked to the increased synthesis of milk fat, which is in line with the results obtained in our study linking grazing with higher milk fat content [45], and conversely, relating higher proportion of concentrate in ration with lower milk fat content (intensive farms) [46].

However, another study showed that when the grain-fed cattle were compared to the grazing cattle, the *bta-miR**-103* content in plasma tended to be higher in the first group (*p* = 0.057), although the difference was not significant [27]. Differences between the previous studies and the present work may reflect differences in the type of tissues and to the fact of studying the cows individually, which may introduce individual variability, such as lactation stage and number of lactations [43]. These and other factors can even mask differences in the expression of miRNAs according to production system [47].

*Bta-miR-155* was abundant in milk from grazing farms compared with corn silage farms. This miRNA was implicated in aspects of energy balance regulation: feed restriction upregulates it in the mammary gland tissue of dairy cows [30], while oxidative stress upregulates it in a mouse model [48]. Negative energy balance in dairy cows activates the production of reactive oxygen metabolites, which in large quantities can create oxidative stress [49]. Grazing may improve immune function and oxidative status [50] due to the high amounts of antioxidants in fresh grass and the exercise involved [51]. On the contrary, other studies have reported the opposite results, with grazing favoring an increase in free radicals without a concomitant increase in the amounts of antioxidants [52]. In grazing cows, the energy input may be lower than in intensive farming [53], which is probably at the origin of the underlying *bta-miR-155* levels in our study.

Studies linked *miR-155* levels to a proinflammatory response in humans [54] and dairy cattle [24]. In dairy cattle, the upregulation of *bta-miR-155* was related to significantly higher risk of mastitis [24]. On grazing farms, it is more likely to have higher levels of somatic cells in milk compared to housed animals [55]; an SSC level above a certain level indicates inflammatory risks, such as mastitis [56], but below these levels may indicate resilience capacity [57]. Therefore, SSC levels may be linked to the differences in *bta-miR-155* levels between grazing or housed cattle.

Furthermore, *bta-miR-7863* is more abundant in zero-grazing farms compared to grazing and corn silage animals. *Bta-miR-7863* has been studied as a mammary biomarker of mastitis caused by *Staphylococcus aureus* and *Escherichia coli* [58].

To our knowledge, no studies have investigated *bta-miR-532* expression in milk so far.

The four miRNAs identified in this study are related to the MAPK pathway, which regulates cell cycle entry and proliferation [59] and the molecular function of serine/threonine-protein kinases, which regulate cell proliferation, programmed cell death (apoptosis), cell differentiation, and embryonic development [60]. Thus, miRNAs *bta-miR-155*, *bta-miR-103*, *bta-miR-532* and *bta-miR-7863* may participate in cell differentiation in the mammary gland and thereby regulate milk production. Our GO-enrichment analysis identified 249 target genes of the 4 studied miRNAs that are related to transport activity, which may indicate the great involvement of the 4 studied miRNAs in the milk synthesis precursor transport process.

Using DIANA miRPath, experimentally validated human targets were identified, such as metabolic pathways related to fatty acid elongation and metabolism for *miR-103* and *miR-155*. This suggests that the high levels of these two miRNAs in cow’s milk are related to the increased fat synthesis due to the consumption of fresh grass, especially during grazing [45]. In parallel this leads us to think about the importance of confirming these results and to study the expression of these miRNA according to the fatty acid profile of milk.

We observed variations in four miRNAs among the four types of farms grouped according to the presence or absence of grazing and different ration ingredients. We did not consider quantitative factors, such as daily grazing hours, pasture management, vegetation, animal density, and amounts of ration ingredients. This could the reason for the large variations in miRNA levels in the groups observed in this study. It would be interesting to investigate, in further studies, the variation of miRNA levels in milk by including these quantitative factors. Within the groups, the farms are not perfect replicates, as the samples belong to commercial farms, which increases the variability. In some groups, we were unable to obtain a large number of samples, as in the case of zero grazing and grass silage. This is due to the fact that few farms adopt this management system. In fact, this is the first investigation of miRNA variation in cow’s milk on commercial dairy farms representing a wide range of production systems.

## 4. Materials and Methods

### 4.1. Study Farms

The sampled farms are located in different parts of Asturias (Spain) and are representative of the characteristic production systems in the north of Spain. For each farm, the following data about feed for lactating cows were requested at three days before site visits: diet composition, whether fresh grass was consumed, and whether grass was consumed as grazed or cut. Data were also collected about the number of lactating cows, breed, and average milk production during the three days prior to the site visit.

### 4.2. Farm Classification

Given the continuum of farm extensiveness and lack of clear regulatory definitions [1], we defined four milk production systems for the present study based on what we considered the most relevant factors and based on the approach of Abou el qassim (2017) [39]. We considered only the management system and the presence or absence of certain ingredients in feed ration. We did not take into account quantitative variables, such as hours spent grazing, or the levels or proportions of certain ingredients in the food ration. In this way, we divided farms into the following four groups (Table 4): grazing (n = 44 farms), where animals had access to grazing and ate fresh grass and concentrated feed (grazing represents the extensive system); zero grazing (n = 13), where animals received a ration of fresh grass and concentrated feed in the stable, without grazing; grass silage (n = 10), where animals received ration based on grass silage and concentrated feed, without grazing or corn silage (zero grazing and grass silage represent intermediate systems); and corn silage (n = 45), where animals received ration of corn silage and concentrated feed, without grazing (corn silage represents intensive system).

### 4.3. Sample Collection and Processing

A total of 112 raw tank milk samples were collected, representing 112 farms, with 10 Holstein cows in the smaller farms and 250 Holstein cows in the bigger one, during fall 2016, spring 2017, and both fall and spring, 2019 and 2021. Samples included two milking sessions: afternoon and morning.

Tubes containing 50 mL of each milk sample were centrifuged at 1900× *g* for 20 min. The fat layer was transferred to fresh 50 mL RNase-free tubes, then Qiazol lysis reagent (Qiagen, Barcelona, Spain) was added (1 mL per milk fat gram). Tubes were vortexed until the fat was thoroughly dispersed, and samples were stored at −80 °C until RNA extraction.

### 4.4. Total RNA Extraction

Total RNA was extracted from 2 mL of milk fat with Qiazol using the mirVana isolation Kit (Ambion) following the manufacturer’s instructions. RNA was eluted in 100 µL of nuclease-free deionized water. The Nano-Drop spectrophotometer (ND-1000, Thermo Fisher Scientific, Madrid, Spain) was used to assess RNA concentration and purity (A260/280 ratio).

### 4.5. Quantitative Real-Time PCR

The isolated total RNA was reverse transcribed using the TaqMan Advanced miRNA cDNA Synthesis Kit (Thermo Fisher Scientific, Madrid, Spain). Levels of miRNAs were determined by quantitative real-time PCR (qRT-PCR) using the TaqMan Advanced miRNA Assay (ThermoFisher Scientific, Madrid, Spain) and a StepOne thermocycler (Applied Biosystems, Foster City, CA, USA) under the following conditions: 95 °C for 20 s, followed by 40 cycles of 95 °C for 1 s and 60 °C for 20 s. Template cDNA (5 μL of a 1:10 dilution) were added to 15 μL of a mix comprising 10 μL of 2× TaqMan Fast Advanced Master mix, 1 μL of 20× TaqMan Advanced miRNA Assay, and 4 μL of RNase-free water.

Specific miRNAs were quantified after selection based on previous sequencing studies [39]. Quantified miRNAs included *bta-miR-215*, *bta-miR-369-5p*, *bta-miR-6520*, *bta-miR-7863*, and *bta-miR-532*, all of which have been identified as the most highly expressed in milk [40]. We also quantified several miRNAs that have been associated with feeding and metabolism: *bta-miR-148*, *bta-miR-155*, *bta-miR-451-5p*, *bta-miR-103*, *bta-miR-181*, *bta-miR-21-5p*, and *bta-miR-29* [27,28,41,61]. The relative abundance of miRNAs was quantified using the △△Ct method after normalization with *bta-miR-30* and *bta-miR-151* as described by Abou el qassim et al., (2022) [40]. All reactions were performed in duplicate. Negative controls lacking cDNA were included in all experiments.

### 4.6. Prediction of Potential Functions and Pathways of Genes Targeted by Milk miRNAs

Targets of miRNAs with different levels across farms were identified using TargetScan 7.2 [62] in the cow database. Target genes were selected based on cumulative weighted context++ score > 0. Functional enrichment analysis of signaling pathways involving the miRNAs target genes was performed using DAVID Bioinformatics tools (v.6.8). The Kyoto Encyclopedia of Genes and Genomes (KEGG) pathways, biological process, and molecular function were analyzed [63].

The targets of the studied milk miRNAs and their associated pathways were also analyzed using DIANA miRPath 3.0 [64]. The validated miRNA targets were identified using DIANA TarBase 7.0 (http://diana.imis.athena-innovation.gr/DianaTools/ accessed on 1 May 2022). Gene Ontology (GO) categories and KEGG were assessed. As bovine genes are not included in DIANA miRPath, target prediction and pathways analysis were performed based on human miRNA annotations.

Statistical analyses were carried out using the integrated Fischer’s exact test followed by FDR (false discovery rate) adjustment.

### 4.7. Statistical Analysis

Data were analyzed using SPSS 22 software. One-way ANOVA was used to compare miRNA levels among the four study groups as independent biological types. When the analysis of variance gave a significant difference for the main effect, the Bonferroni post hoc test was applied (multiple comparison of means). When the assumption of equal variances was not met, Welch’s ANOVA was used as an alternative to one-way ANOVA. When the normality of the residues and homogeneity of variance was not verified, means were compared using the Kruskal–Wallis non-parametric test; when significant differences were attained, the Games–Howell post hoc test (multiple comparison of means) was performed. Significance was defined as *p* ≤ 0.05.

## 5. Conclusions

This study confirms that the dairy production system could influence miRNA levels in milk fat. In particular, the miRNAs *bta-miR-103* and *bta-miR-155* are significantly abundant in grazing farms and may be related to different factors intrinsic to grazing, such as diet and fresh grass consumption, as well as exercise and other aspects, such as immune response and oxidative status. These miRNAs emerge as potential biomarkers for the tracing of pasture-based milk.

## Figures and Tables

**Figure 1 ijms-23-11681-f001:**
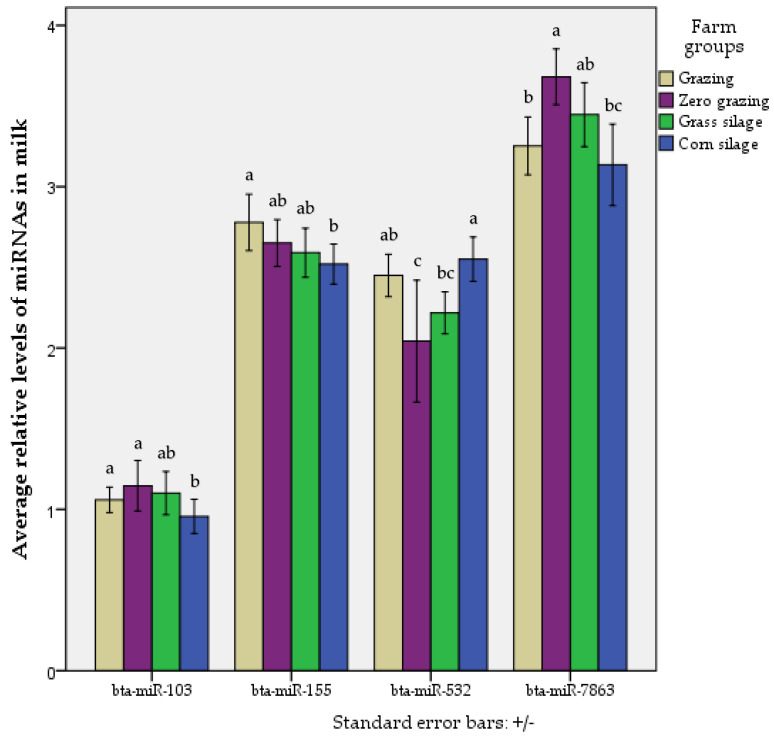
Average relative levels of the miRNAs *bta-miR-103*, *bta-miR-155*, *bta-miR-532*, and *bta-miR-7863* in raw milk from grazing (n = 44), zero-grazing (n = 13), grass silage, (n = 10), or corn silage (n = 45) milk production systems. The bar chart shows the average of miRNA levels in each farm group, and the standard error bars. Different letters show significant difference between groups.

**Figure 2 ijms-23-11681-f002:**
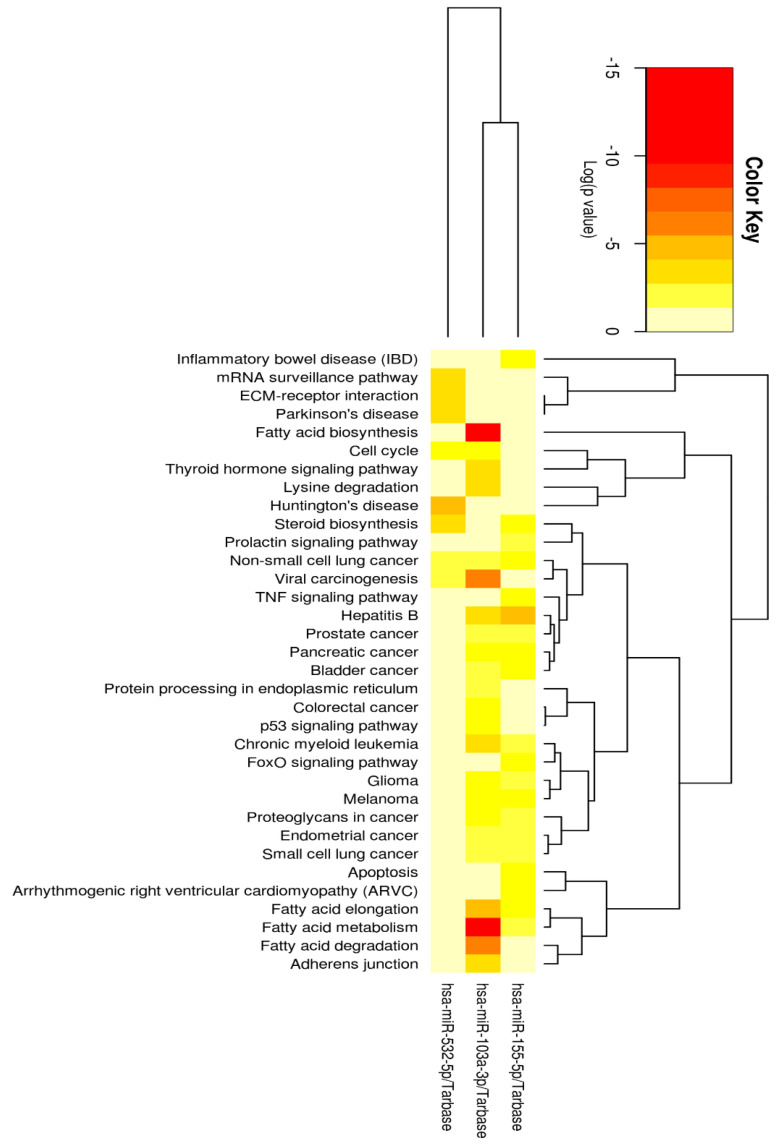
Heatmap of hierarchical clustering of *miR-103*, *miR-155* and *miR-532* based on mRNA target pathways, identified in DIANA using the Tarbase and KEGG pathway union representation. Darker colors represent lower *p*-values.

**Table 1 ijms-23-11681-t001:** KEGG pathways that are associated with milk production and metabolism and that are predicted to be regulated by milk miRNAs with differential levels across production systems.

KEGG Signaling Pathway	*Biomarker*				No. of Target Genes	*p*-Value
	*bta-miR-103*	*bta-miR-155*	*bta-miR-532*	*bta-miR-7863*		
*AMPK ^1^*	x	x		x	38	4.6 × 10^−7^
*MAPK*	x	x	x	x	67	3.2 × 10^−6^
*PI3K-Akt ^2^*		x	x	x	27	5.6 × 10^−5^
*Oxytocin*	x	x		x	38	1.1 × 10^−4^
*Prolactin*		x		x	25	1.2 × 10^−4^
*Insulin*		x		x	35	1.4 × 10^−4^
*Ras ^3^*	x	x		x	53	1.6 × 10^−4^
*Growth hormone synthesis*, *secretion and action*		x		x	31	1.8 × 10^−4^
*TGF-beta ^4^*		x		x	25	6.2 × 10^−4^
*Calcium*	x			x	52	7.3 × 10^−4^
*Glucagon*				x	24	6.0 × 10^−3^
*Lipid and atherosclerosis*	x	x			43	2.5 × 10^−2^
*cGMP-PKG ^5^*	x				32	3.3 × 10^−2^
*Mineral absorption*				x	13	8.3 × 10^−2^
*Lysine degradation*				x	14	9.7 × 10^−2^

^1^ Adenosine monophosphate-activated protein kinase (AMPK), ^2^ phosphoinositide 3-kinases- protein kinase B (PI3Ks-Akt), ^3^ rat sarcoma virus (Ras), ^4^ transforming growth factor beta (TGF-beta), ^5^ cyclic guanosine monophosphate- protein kinase G (cGMP-PKG). x implies pathway associated.

**Table 2 ijms-23-11681-t002:** Biological processes that are associated with milk production and metabolism and that are predicted to be regulated by milk miRNAs with differential levels across production systems.

Biological Process	*Biomarker*				No. of Target Genes	*p*-Value
	*bta-miR-103*	*bta-miR-155*	*bta-miR-532*	*bta-miR-7863*		
*Growth regulation*				x	11	2.5 × 10^−3^
*Transport*				x	249	1.4 × 10^−2^
*Ion transport*				x	79	1.6 × 10^−2^
*Calcium transport*				x	15	1.7 × 10^−2^
*Amino- acid transport*				x	7	2.4 × 10^−2^
*Protein transport*	x			x	67	2.7 × 10^−2^
*Differentiation*		x			45	6.0 × 10^−2^
*Sodium transport*	x				13	9.5 × 10^−2^

x implies biological processes associated.

**Table 3 ijms-23-11681-t003:** Molecular functions that are associated with milk production and metabolism and that are predicted to be regulated by milk miRNAs with differential levels across production systems.

Molecular Function	*Biomarker*				No. of Target Genes	*p*-Value
	*bta-miR-103*	*bta-miR-155*	*bta-miR-532*	*bta-miR-7863*		
*Transferase*	x	x	x	x	279	1.4 × 10^−20^
*Activator*	x	x		x	64	1.3 × 10^−8^
*Ion channel*	x			x	65	3.0 × 10^−8^
*Serine/threonine-protein kinase*	x	x	x	x	55	7.1 × 10^−8^
*Developmental protein*	x	x		x	64	3.0 × 10^−6^
*Glycosyltransferase*				x	31	2.5 × 10^−3^
*Hydrolase*	x		x	x	182	2.8 × 10^−3^
*Growth factor*				x	17	9.0 × 10^−3^
*Protein phosphatase*			x	x	19	1.3 × 10^−2^
*Guanine-nucleotide releasing factor*		x		x	16	1.6 × 10^−2^
*Calcium channel*				x	8	4.0 × 10^−2^
*Potassium channel*				x	12	4.8 × 10^−2^

x implies molecular functions associated.

**Table 4 ijms-23-11681-t004:** Classification of the farms in the study based on milk production system.

	Ration Composition				
Production System	Grazing	Fresh Grass in the Stable	Grass Silage	Corn Silage	Concentrated Feed
Grazing (n = 44)	+	-	-	-	+
	+	+	-	-	+
	+	-	+	-	+
	+	-	+	-	+
	+	-	-	-	+
	+	+	+	-	+
	+	-	+	+	+
	+	-	+	+	+
Zero grazing (n = 13)	-	+	+	-	+
	-	+	-	-	+
	-	+	+	-	+
	-	+	-	-	+
Grass silage (n = 10)	-	-	+	-	+
Corn silage (n = 45)	-	-	+	+	+

+ implies presence in the diet, - implies no presence in the diet.

## Data Availability

The data that support the findings of this study are available from the corresponding author upon reasonable request.

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
