# Peer review of "Variation of miRNA Content in Cow Raw Milk Depending on the Dairy Production System"

_ijms, 2022, doi:10.3390/ijms231911681_

Round 1

Reviewer 1 Report

The manuscript of Abou el qassim et al., is a well-written research manuscript presenting novel data on cow miRNAs modulated by dietary habbits and the potential use of these differentiated miRNAs as biomarkers of milk quality. Indeed, the authors explore those differentially expressed miRNAs and they explore by bioinformatics their potential biological effects. By contrast, the paper's objective needs to be clearly outlined and addressed (needs to improve the biomarker part and enhance the potential of bioactivity).

Introduction.

In this part, authors need to outline that bovine miRNAs could not only act as a biomarker of might possess biological properties (i.e. Tomé-Carneiro et al., 2016, del Pozo et al., 2021… ) In general, Davalos’s group or Zempleni’s group publications may help to address this topic).

In reference 33 you can use instead a reference from bovine milk miRNAs. There are papers covering this topic.

L 66 Authors should describe the rationale of miRNA’s selection

Results:

Is there a differences on total RNA content depending on milk production system? The authors might include a figure in supplementary material if they found something interesting or a tendency depending on the dietary group or even if milk ripening is affecting to total RNA content.

Figure 1: please identify if axis is relative miRNA expression. It is unclear whether the authors use DDCT how they use the calibrator to adjust one group as control (one of the groups should stay in 1).  Indeed,  a better editing might be clearer. For instance, the four miRNAs can enter in the same line… They are using too much space and it difficults the figure understanding.

I’m not confident that the selected miRNAs might serve as biomarkers of compliance… I think that authors should explore deeply their potential… They mentioned in M&M section that samples were acquired in fall 2016, spring  2017, and both fall and spring 2019 and 2021. There are differences between each period in terms of miRNA expression?

The authors checked if the samples were maintained in good conditions?

It could be also interesting to test the expression profile of miRNAs evolution during the lactation period. And check if the miRNA expression profile affects to discrimine if this biomarker could be robust.

Authors should validate in a large cohort if the proposed biomarkers are robust and could be used as candidates.

Include in supplementary material those miRNAs that not respond to production system.

Tables: use . instead of , in the p-value column.

Table 2 amino-acid?

The selected miRNAs shows high/low homology with those hsa-miRNAs? Check and include.

Figure 2 seems a bit our of place here; If the paper is focus on identifying potential biomarkers to identify milk from different dairy production systems, the biological effects should be removed. Otherwise, authors should reformulate the hypothesis of the manuscript (and title, and include cross-species hypothesis and evidences, etc).

Authors should explore/validate by in vitro methods the results obtained by bioinformatics.  

Discussion:

This study contains several limitations that should be described. Please address this topic.

In this part… authors should discuss if miRNAs could be maintained during GI conditions and produce the effects described in paragraph L203.

Material and methods:

L:265 include the reference

Conclusions:

L305: this sentence is too risky with the proposed experiments. Authors should be aware of this conclusion. They should increase the experiments to confirm a solid modulation of miRNAs to conclude it.

Supplementary material: This folder is broken and not readable. I cannot open it. So I cant’ comment on it.

Author Response

Response to Reviewer 1 Comments

I would like to thank you for taking the time to read and comment on our article. In the following manuscript, we will answer your questions and clarify several statements that were ambiguous than intended and adjust the text to be clearer. Hereafter We will comment point by point

The manuscript of Abou el qassim et al., is a well-written research manuscript presenting novel data on cow miRNAs modulated by dietary habbits and the potential use of these differentiated miRNAs as biomarkers of milk quality. Indeed, the authors explore those differentially expressed miRNAs and they explore by bioinformatics their potential biological effects. By contrast, the paper's objective needs to be clearly outlined and addressed (needs to improve the biomarker part and enhance the potential of bioactivity).

Our main objective is to identify miRNA in milk that could differentiate grazing production systems, then we carried out bioinformatic analysis to try to find out the metabolic pathways in which the differentiating miRNAs are involved and to try to explain the results of the differential miRNA amounts between farm groups, from a biological point of view. Studying, the potential of bioactivity of miRNA in milk It is quite an interesting topic that we are not exploring in this paper, however it would be interesting to assess the functions of miRNAs with high levels in grazing milk to highlight the functional role of these miRNAs for consumer (Further explanations in the replies to comments)

see L100

  • In this part, authors need to outline that bovine miRNAs could not only act as a biomarker of might possess biological properties (i.e. Tomé-Carneiro et al., 2016, del Pozo et al., 2021… ) In general, Davalos’s group or Zempleni’s group publications may help to address this topic).
  • This part was added to highlight the bioactive characteristic of milk in the line 53 “Beyond their functions in the cells that produce them, microRNAs can also be transferred to other cells, or to other species, in protein complexes or through extracellular vesicles (Zhang et al., 2015; Zhou et al., 2017). There is also compelling evidence that humans use microRNAs from cow's milk in gene regulation (Zempleni et al. 2015; Del Pozo-Acebo et al., 2021), highlighting the bioactive characteristic of milk.”
  • Zhang, J., Li, S., Li, L., Li, M., Guo, C., Yao, J., & Mi, S. (2015). Exosome and exosomal microRNA: trafficking, sorting, and function. Genomics, proteomics & bioinformatics13(1), 17-24.
  • Zhou, G., Zhou, Y., & Chen, X. (2017). New insight into inter-kingdom communication: horizontal transfer of mobile small RNAs. Frontiers in microbiology8, 768.
  • Zempleni, J., Baier, S. R., Howard, K. M., & Cui, J. (2015). Gene regulation by dietary microRNAs. Canadian journal of physiology and pharmacology93(12), 1097-1102.
  • Del Pozo-Acebo, L.; de Las Hazas, M.-C.L.; Tomé-Carneiro, J.; Gil-Cabrerizo, P.; San-Cristobal, R.; Busto, R.; García-Ruiz, A.;
    Dávalos, A. Bovine Milk-Derived Exosomes as a Drug Delivery Vehicle for miRNA-Based Therapy. Int. J. Mol. Sci. 2021, 22, 1105.

  • In reference 33 you can use instead a reference from bovine milk miRNAs. There are papers covering this topic.
  • Reference 33 remplacer by: Line 69
  • Izumi, H.; Kosaka, N.; Shimizu, T.; Sekine, K.; Ochiya, T.; Takase, M. Bovine Milk Contains MicroRNA and Messenger RNA That Are Stable under Degradative Conditions. J. Dairy Sci. 2012, 95, 4831–4841, doi:10.3168/jds.2012-5489.

  • L 66 Authors should describe the rationale of miRNA’s selection
  • Change done: Hence, in the current study, the quantification of the abundance of 12 selected miRNAs was performed in tank milk from four dairy farming systems: grazing, zero-grazing, grass silage or corn silage. Several miRNAs were selected from a previous sequencing work (Abou el qassim 2017) and others were selected from the literature for being associated with feeding and metabolism. L72

You can find more details in materials and methods

  1. Results:
  • Is there a difference on total RNA content depending on milk production system? The authors might include a figure in supplementary material if they found something interesting or a tendency depending on the dietary group or even if milk ripening is affecting to total RNA content.
  • We did not find any differences on total RNA content or quality depending on milk production system

Significance table of the differences in total RNA between the different farm groups

zero grazing

grazing

corn silage

grass silage

zero grazing

grazing

0,641642

corn silage

0,853491

0,555653

grass silage

0,616058

0,869354

0,615965986

  • Figure 1: please identify if axis is relative miRNA expression.
  • Yes, the axis is relative miRNA expression. Changed in the text

  • It is unclear whether the authors use DDCT how they use the calibrator to adjust one group as control (one of the groups should stay in 1).
  • we usually use Q base + software: we referred all the samples to the smallest sample value, (there are more options: refer to the max, to the average, to a random sample). Any way it gives the same result, our choice has been made to have the results in positive and to be able to compare the averages. The internal normalization is based on the expression of miRNAs with stable expression between the different farms and which have been determined in the work (Abou el qassim 2017). Using the GeNorm algorithm.

How we carried out the normalization?

Normalization is an essential component of a reliable qPCR assay. geNorm is one of the most popular algorithms to find stable reference genes from a set of tested candidate reference genes in a given experimental condition.

First, from the sequencing study we chose those miRNAs with more stable expression among samples in all production systems (miRNA that do not vary according to production system), so we selected the miRNAs with the smallest coefficient of variation (CV= standard deviation / mean).

We used geNorm to find the optimal number and choice of miRNA for normalization, validating by qRT-PCR miRNAs identified in sequencing analysis as stable in 22 tank milk samples representing all the experimental variation of dairy production systems existing in the area of study, in our case bta-miR-30 and bta-miR-151. So we use a normalization factor  for each sample based on the geometric mean of bta-miR-30 and bta-miR-151.

  • Indeed, a better editing might be clearer. For instance, the four miRNAs can enter in the same line… They are using too much space and it difficults the figure understanding.
  • Each miRNA is independent from the others, the most interesting information from the figure is the comparation of the expression of the different miRNA among groups.

  • I’m not confident that the selected miRNAs might serve as biomarkers of compliance… I think that authors should explore deeply their potential… They mentioned in M&M section that samples were acquired in fall 2016, spring 2017, and both fall and spring 2019 and 2021. There are differences between each period in terms of miRNA expression?
  • 6 miRNAs are selected on the basis of a sequencing study of two groups of milk from very extensive (corresponding to the grazing group in this case) and very intensive (corresponding to the corn silage group) dairy farms. The sequencing results of these miRNAs showed significant differences between the two extremes of milk production systems. They are candidate miRNAs that can change according to changes in diet and animal management. Furthermore, we have selected more miRNAs from other studies (the literature) that showed that these miRNAs vary according to the exercise (relative to the movement that animals make during grazing), feeding and try to validate them in our conditions.

During each sampling (fall 2016, spring 2017, and both fall and spring 2019 and 2021), the farms were from different production systems to dilute the effect of year and season, all sampling period include samples from all groups. We have 112 samples, whose PCR have been performed on 4 plates. As a plate can be viewed as a statistical block, so, we had randomized the four treatments in the 4 plates. Then we used inter-run calibrators on each plate to correct plate-to-plate variation. In other words, in each group of production systems (samples from fall 2016, spring 2017, and both fall and spring 2019 and 2021 were included.

  • The authors checked if the samples were maintained in good conditions?
  • All milk samples were collected from tank milk in commercial farms by ourselves and kept cold in a refrigerator during transport to the laboratory. The time between the collection of the sample and its processing in the laboratory never exceeds 3 hours. On arrival at the laboratory, the fat was separated by centrifugation and kept in the QIazol at -80ºC until we get all the samples of the season (e.g. fall 2017) for further processing. Once we have the samples from each period, maximum one month after the end of sampling, total RNA extraction was done and RNA stored immediately at -80ºC until use for quantity and quality measure and cDNA synthesis. So, although the sampling has been done in different years, the samples have been processed close to the sampling and always maintained at -80ºC. To plate to plate calibration (inter run calibration), one sample is always run in all the plates, plus one sample from the previous plate, helping to set the same threshold for all plates

  • It could be also interesting to test the expression profile of miRNAs evolution during the lactation period. And check if the miRNA expression profile affects to discrimine if this biomarker could be robust.
  • We are working with tank bulk milk, so all individual effects are diluted. We are interested in the whole farm system. In commercial dairy farms animal reproduction programs are usually stablished to have lactating cows in different lactating periods, to avoid as much differences as possible in milk production during the year.

  • Authors should validate in a large cohort if the proposed biomarkers are robust and could be used as candidates.
  • The levels of 12 miRNAs are validated in 112 farms.

As mentioned by Udvardi et al. (2008), an experiment ideally should encompass at least three independent biological replicates of each treatment(farm groups).

Ideally, a full experiment would be analyzed on a single (typically 96-well) plate. However, an experiment with many treatments requires a design strategy for multiple plates. We have 112 samples, whose PCR have been performed on 4 plates. As a plate can be viewed as a statistical block, so, we had randomized the four treatments in the 4 plates. Then we used inter-run calibrators on each plate to improve the assessment of plate-to-plate variation.

For each biological replicate, it is common to run at least two technical replicates of each PCR reaction (Ivo Rieu and Stephen, 2009). To perform the qRT-PCR, every sample was used in replicate. So, when we got the results on the q Base + (the used platform for expression analysis) only the duplicates with a ct less than 0.5 were maintained.

With a good randomization of the treatments between the plates, the use of the calibrator between the plates and working with the replicas as similar as possible we believe that our estimations are valid.

  • Include in supplementary material those miRNAs that do not respond to production system. Tables: use . instead of , in the p-value column.
  • The proposed change is done.

  • Table 2 amino acid?
  • Sorry I do not get what is the question, here we mean amino acid Transport biological process

  • The selected miRNAs shows high/low homology with those hsa-miRNAs? Check and include.
  • Added in the L130
  • There are few platforms for the study of bovine functionality analysis. Human annotations are often used because they have more information. Before using Diana miRpath we have tested the similarity between human and cow sequences.

The bovine and human miRNA sequences from the miRbase platform:

>hsa-miR-155-5p MIMAT0000646

UUAAUGCUAAUCGUGAUAGGGGUU

>bta-miR-155 MIMAT0009241UUAAUGCUAAUCGUGAUAGGGGU

>bta-miR-103 MIMAT0003521AGCAGCAUUGUACAGGGCUAUGA>hsa-miR-103a-3p MIMAT0000101AGCAGCAUUGUACAGGGCUAUGA

>hsa-miR-532-5p MIMAT0002888CAUGCCUUGAGUGUAGGACCGU>bta-miR-532 MIMAT0003848CAUGCCUUGAGUGUAGGACCGU

Bta-miR-7863 does not exist in human data-base, so it has been analysed only with targetScan.

  • Figure 2 seems a bit out of place here; If the paper is focus on identifying potential biomarkers to identify milk from different dairy production systems, the biological effects should be removed. Otherwise, authors should reformulate the hypothesis of the manuscript (and title, and include cross-species hypothesis and evidences, etc). Authors should explore/validate by in vitro methods the results obtained by bioinformatics.  
  • The aim of the bioinformatic analysis here is to try to find out the metabolic pathways in which the differentiating miRNAs are involved and to try to explain the results of the differential miRNA amounts between farm groups, from a biological point of view. Due to the lack of bovine bioinformatic tools, we have used database platforms based on human annotations, after verifying the sequence similarities of these miRNAs between Bos taurus and humans in miRBASE platform. We used target Scan in bovine which gave us some information that was completed with miRpath (human annotations). In several articles studying the expression of miRNAs in cow's milk, the same methodology has been followed:
  • Ioannidis, J., & Donadeu, F. X. (2017). Changes in circulating microRNA levels can be identified as early as day 8 of pregnancy in cattle. PloS one, 12(4), e0174892.
  • Saenz-de-Juano, M. D., Silvestrelli, G., Bauersachs, S., & Ulbrich, S. E. (2022). Determining extracellular vesicles properties and miRNA cargo variability in bovine milk from healthy cows and cows undergoing subclinical mastitis. BMC genomics, 23(1), 1-15.
  • Webb, L. A., Ghaffari, M. H., Sadri, H., Schuh, K., Zamarian, V., Koch, C., ... & Sauerwein, H. (2020). Profiling of circulating microRNA and pathway analysis in normal-versus over-conditioned dairy cows during the dry period and early lactation. Journal of dairy science, 103(10), 9534-9547.
  • Le Guillou, S., Marthey, S., Laloë, D., Laubier, J., Mobuchon, L., Leroux, C., & Le Provost, F. (2014). Characterisation and comparison of lactating mouse and bovine mammary gland miRNomes. PloS one, 9(3), e91938.

Authors should explore/validate by in vitro methods the results obtained by bioinformatics.  

The suggested experiment is interesting and would provide additional information about..., but we feel that it falls outside the scope of this study.

Discussion:

  • This study contains several limitations that should be described. Please address this topic.

L221

  • We observed variations in four miRNAs among four types of farms grouped according to the presence or absence of grazing and different ration ingredients. We did not consider quantitative factors, as daily grazing hours, pasture management, vegetation, animal density, and amounts of ration ingredients. This could the reason for the large variations in miRNA levels in the groups observed in this study. It would be interesting to investigate, in further studies, the variation of miRNA levels in milk by including these quantitative factors.

  • Within the groups, the farms are not perfect replicates, as the samples belong to commercial farms, which increases the variability.

  • In some groups, we were unable to obtain a large number of samples, as in the case of zero grazing and grass silage. This is due to the fact that few farms adopt this management system.

Lines 221 to 232

  • In this part… authors should discuss if miRNAs could be maintained during GI conditions and produce the effects described in paragraph L203.
  • In the results section we presented the results of PCRs on one hand and the main functions, related to metabolism … on the other hand. In the discussion we tried to explain the levels of miRNAs among groups, citing some studies treating the same point and then explaining the levels from a biological point of view through the functional analysis results.

  • we linked bta-miR-155 with pro-inflammatory functions (bioinformatics analysis) and then we mentioned that in the grazing production system it is more likely to have higher levels of somatic cells, which indicates up to a certain level inflammatory risks such as mastitis giving a possible explanation for the level of bta-miR-155 in milk from grazing farms.
  • in the case of bta-miR-103 we found a greater levels in the groups grazing and 0 Grazing comparing to the rest. The variables in common between these two types of farms is the consumption of fresh grass. we related the expression of bta-miR-103 with stearoyl-CoA desaturase (SCD) linked to increased synthesis of milk fat, at the same time we linked the increase of fat synthesis in milk with the consumption of fresh grass consumption demonstrated in several works
  •  
  1. Material and methods:

  • L:265 include the reference
  • (Abou el qassim, 2017) L 277

  1. Conclusions:

  • L305: this sentence is too risky with the proposed experiments. Authors should be aware of this conclusion. They should increase the experiments to confirm a solid modulation of miRNAs to conclude it.
  • In this experiment we use a large number of farms, randomising the effects of year and time of sampling. Individual animal effects are diluted, because the milk is sampled from the tank representing all lactating animals, so the most likely main effect that may be behind variations in miRNA levels is the production system, which could be the sum of the intrinsic factors of the production system. The methodology in the laboratory has been performed in the most optimised way possible as described above.
  • This study confirms that the dairy production system could influence miRNA levels in milk fat.

Then we tried to introduce some explanation about the possible raison behind the levels of miRNA between the groups of farms.

We mentioned the intrinsic factorsof production system that may be behind the variation of miRNA levels (grazing, such as diet and fresh grass consumption, as well as exercise through grazing, because in this systems cows walk more). In addition to this we tried to introduce other possible explanations based on the consequences of the production systems (for eg: intensive systems can be more related with stressed animal) that is why we mention oxidative status, and also because the bioinformatic analysis showed that some of these miRNAs are implied in this processes.

In particular, the miRNAs bta-miR-103 and bta-miR-155 are significantly abundant in grazing farms and may be related to different factors intrinsic to grazing, such as diet and fresh grass consumption, as well as exercise and other aspects like immune response and oxidative status.

These miRNAs emerge as potential biomarkers for the tracing of pasture-based milk.

Supplementary material: This folder is broken and not readable. I cannot open it. So I cant’ comment on it

The proposed change is done

Thank you for your reflexions

Reviewer 2 Report

Dear authors,

Your article made in the first glance a good impression on me. In reading it, some major questions arose, which should be clarified.

1)      DATA origin and statistics: you used 112 samples – of how many farms? When I calculate the several sampling dates, it may be 6 samples per farm. Was this considered in the statistics? (I think, it was not. Why?).

      2)      Measuring validity: You did use two replicates – are you sure that measuring variation is low in relation to sample variation?

      3)      Discussion and conclusion: You showed the different functions of the miRNA’s – but in Line 194-195 you say that all four miRNA’s are involved in cell proliferation. But I see different level for the farming systems for the miRNA’s, thus different pattern, and why can they in this case indicate all the same physiological state? I feel as if the discussion part should be improved a lot – to give the reader a clear impression what do you think about your results, in relation to the farming system (it would be fine to explain WHY the grazing system is the extensive one, and not only refer to “previous work” – and to link the systems with functions/pathways likely to be present under the given feeding regime).

Remarks in detail:

L34: a space before [5] is missing.

L37: a space before [4,6] is missing

L89-92 (Figure 1): An “overall mean-line” could help to identify higher or lower level.

If the layout of the paper allows, it would be fine to see the four graphs on one page, arranged as 2x2 – to see the different level-pattern. I see in them two pattern: Pattern1 = RNA155, with decreasing from left to right side arranged system (from extensive to intensive system), and Pattern2 = 103, 7863 (and reversely 532), with higher amounts for the first and last system (in tendency), and contrary to the zero-grazing and grass-silage system. It could be interesting to try to explain this!

L92: In the Figure of 7863, please check the letters for significance: in general, the highest value is represented by “a”. Why did you use “a” for the low-level grazing value? (the same for 532).

L102, L111: KEGG and MAPG are abbreviations – are they as popular that an explanation is not needed? What about GO? (L126)

Table 1, 2, ff: The font size in the tables looks as over-sized in relation to the text – please check if this is correct! It would be better to see all information “at one glance” if the font is smaller and the space in-between the words / symbols is less.

The header of the table should be repeated with a page break – but it would be fine to avoid this!

L124: please repeat “genes” in the brackets for miRNA155 and miRNA532

Please check for consistent names for the miRNA’s (miR-xxx or mi-RNA-xxx or bta-miRxxx?)

Why is the miR-103 added with a-3p?

L129: you tell that the miR155 is involved in FA elongation – but in the Figure 2 the heatmap is light – is the supplementary Figure 1 telling why 155 is involved in FA elongation?

L147: why did you not show miR215 of the actual data? (it is not significantly different, I know. But you want to link to previous work – this given statement is not much worth if no comparison is given to present data!

L149: here you could explain why grazing is extensive.

L150ff: The question arose: in which case is miR103 different? On which factor does it react? It would be fine to relate the observations to functions – to draw a picture for the reader how to understand the observations.

L162: “in” is missing : “which is IN line with…”

L162: “an” is missing: “Howerver, ANother study…”

-Maybe you want to structure your Discussion to “functions of miR103”, “functions of miR155” etc? In the results section, you tell much more about functions as in the discussion – it would be fine to find here more explanations of functions in relation to systems.

L178: Fresh grass was fed also in barn (stable). But in barn, frequently a higher amount of concentrates is fed, this result in “more intensive” systems (higher milk yield) – why did you not try to evaluate the data in relation to milk yield as a separate factor to the actual systems? I think, the amount of concentrated forages is of importance, not whether the cows graze on the pasture or eat it in the barn (maybe due to long ways to the pasture? Or to reduce feeding-losses, because cut grass is eaten for up to 100%, while grazed areas have losses by the steps of the cows)

L187: “inflammatory response = SCC high” : did you ever think on the fact that a healthy organism is regularly re-newing it’s tissue? And a good health may also be the ability to fight against infections – by tissue recovery (resulting in higher SCC). This may be named “resilience”.

L213: I think the sentence has to be like this: “…this could be THE REASON FOR the LARGE variation…”

L223: the fresh grass AMOUNT would be very fine to know, at least a relative measure (approx.. 30% of the diet, 80% of the diet) or so.

L230: I would be glad to read here an explanation WHY this grouping was used, not only the reference.

L231: WHY did you think that the quantitive variables are unimportant? I think they are very much of importance!!!

Table 4: could you add the number of samples for each system? (n=xx)

Think of repated header if there is a page break.

L242: please give the number of farms, and the number of samples per farm. Did you reflect this repeated measures in the statistics?

L246: how did you manage the transport? (time, cooling system?)

L272: you performed duplicate measures. How did you evaluate measurement uncertainty?

L287: you mention the Fischer’S test – on which data level did you use it? On raw date? With which aim? How did you manage the dublicate data?

L321ff – References: some references look as if the given information is not complete – [1], [31], [49], [57] ,  [55] – here it is “master thesis”, an “tesis de master”

In general, I think it is very interesting to look in the farming system effects – but please ensure that the statistics are correctly performed, and the measurements represent valid values!

Good chance!

Round 2

Reviewer 1 Report

Thanks for this extensive an clear review. Now the manuscript can be published. Congratulations. 

Author Response

Thank you very much for your time and comments!

Reviewer 2 Report

Please find the detailed comments in the document added.

Author Response

Response to Reviewer 2 Comments

I would like to thank you again for your time and comments Hereafter We will comment point by point

Line 156: Which of the investigated systems are “extensive” which degree are they extensive

In the same sentence in the line 156 we indicate it “extensive grazing production system”

Also we added in the section materials and methods which group is extensive and which one is intensive

Line 162 uniformity miR-103 bta-miR-103

When we mention bta-miR-103 we refer to cow miRNA

When we mention miR-103 we refer to the results of miRNA from diana which is based on human annotations Line 132, 138, 140, 148,215

In the paragraph from the line 168 to 172 we are mentioning a study based on goat miRNA, so we mentioned also miR-103

Line 177: corrected

Line 198 commentary on the levels of 155 and inflammation

we mentioned that miR-155 at high levels is related to risk of mastitis, that grazing is related to high levels of somatic cells. High levels of somatic cells can show either that cows have mastitis (if the level is over 200,000 cells/ml) or that they have high resilience (if the level is below 200,000 cells/ml).

in the sampled farms the cows would not have mastitis because the milk is intended for consumption. So it is most likely that cows on pasture are more exposed to more pathogen diversity, that makes them stay in a certain immunity level.

The high level of miR-155, at least in our case, would not mean that the cows have mastitis, but may rather be related to a high level of SCC without suffering from mastitis.

The other corrections are done in the same manuscrit.

Thank you for your reviewing, I hope now is ready to be published in the IJMS journal.